# Cooperative Transmission Mechanism Based on Revenue Learning for Vehicular Networks

Mingyang Chen [1,2], Haixia Cui [1,2,*], Mingsheng Nie [1,2], Qiuxian Chen [1,2], Shunan Yang [1,2], Yongliang Du [3] and Feipeng Dai [1,2]

1   School of Electronics and Information Engineering, South China Normal University, Foshan 528225, China
2   School of Physics and Telecommunication Engineering, South China Normal University, Guangzhou 510006, China
3   China Shift Internet Co., Ltd., Guangzhou 510000, China
*   Correspondence: cuihaixia@scnu.edu.cn

**Abstract:** With the rapid development of science and technology and the improvement of people's living standards, vehicles have gradually become the main means of travel. The increase in vehicles has also brought about an increasing incidence of car accidents. In order to reduce traffic accidents, many researchers have proposed the use of vehicular networks to quickly transmit information. As long as these vehicles can receive information from other vehicles or buildings nearby in a timely manner, they can avoid accidents. In vehicular networks, the traditional double connection technique, through interference coordination scheduling strategy based on graph theory, can ensure the fairness of vehicles and obtain suitable neighborhood interference resistance with limited computing resources. However, when a base station transmits data to the vehicular user, the nearby base station and the vehicular network user may be in a state of suspended communication. Thus, the resource utilization of the above double connection vehicular network is not sufficient, resulting in a waste of resources. To solve this issue, this paper presents a study based on earnings learning with a vehicular network multi-point collaborative transmission mechanism, in which the vehicular network users communicate with the surrounding collaborative transmission. We use the Q-learning algorithm in the reinforcement learning process to enable vehicular network users to learn from each other and make cooperative decisions in different environments. In reinforcement learning, the agent makes a decision and changes the state of the environment. Then, the environment feeds back the benefit to the agent through the related algorithm so that the agent gradually learns the optimal decision. Simulation results demonstrate the superiority of our proposed approach with the revenue machine learning model compared with the benchmark schemes.

**Keywords:** vehicular networks; cooperative transmission; reinforcement learning; interference coordination; driverless technology

## 1. Introduction

In recent years, with the increasing improvement of living standards, people's demand for transportation tools has gradually increased, and the number of civil vehicles has also been increasing. By the end of 2020, the number of civil cars in China reached 280 million, and the mass popularization of cars has caused increasingly serious urban traffic jams. The ensuing traffic safety issues have attracted people's attention. Even though the government has come up with new laws and regulations to regulate drivers' driving behavior, there are nearly 200,000 traffic accidents in China every year, resulting in a large number of casualties and property losses. The development of national road traffic should not only pay attention to the quantity and ignore the quality. While realizing the improvement of people's living standard, we should also deal with the social hidden dangers brought by it. Given driver problems, manual driving can reduce some dangers for traffic safety. Therefore, in order to

solve the traffic safety hazards caused by the increase in vehicles, people put their eyes on more intelligent and standardized management of unmanned driving technology [1].

Driverless technology, which allows drivers to safely free their hands and reduce the incidence of traffic accidents, has always been a key direction in the field of automotive research. Domestic scholars' research on unmanned driving technology started in 2014, while foreign research on it started as early as 2004. Although domestic and foreign experts and scholars have made many major breakthroughs in unmanned driving technology and 5G communication technology has become a new driving force for its development, the key to whether unmanned driving technology can be fully utilized lies in whether the communication capacity of vehicle-mounted networks can be improved. Without a high level of in-vehicle communication capability, the adoption of driverless technology will also be difficult. The vehicular network is an important module for building intelligent transportation, and the successful construction of a vehicular network is of great significance in solving traffic congestion and reducing the incidence of traffic accidents. Therefore, the improvement of the communication capability of vehicle networks is the key field of vehicle network technology research. Cooperative transmission technology is the breakthrough to improve the communication technology of vehicular networks. The application of multi-point cooperative transmission technology to the communication of vehicular networks can effectively save the transmission time of the vehicular network, improve the transmission efficiency of data and signals as much as possible, and reduce the probability of transmission errors in the vehicular network and the loss of the wrong frequency and spectrum.

In the traditional communication and transmission network, all users can only receive the signal transmitted by a base station, which is a single point transmission. One of the key technologies of LTE-A is the cooperative service of relay selection with the help of neighboring vehicle network users. By coordinating the information of multiple base stations and forming a cooperative overlapping area of base stations to transmit signals to terminals, the communication efficiency can be improved. The application of multi-point cooperative transmission technology [2] in vehicular networks can transmit signals to edge terminals by using adjacent vehicles as relay nodes [3] and occupy communication resources originally belonging to other vehicular network terminals. The connectivity of a vehicular network can be greatly improved by adding adjacent vehicles to the communication of the vehicular network in the form of relay nodes. The selection of relay nodes determines the performance of the multi-point cooperative transmission system. Therefore, how to better coordinate transmission relay node selection has become the focus of attention. Nowadays, with the rapid development of science and technology, various kinds of algorithms emerge in an endless stream. Machine learning is widely used in the field of artificial intelligence due to its characteristics of constantly learning to acquire new knowledge or ability and reconstructing the existing knowledge structure to continuously improve its performance. Reinforcement learning, as one of the techniques of machine learning, was first proposed by Minsky in 1954. It is a kind of continuous trial training between agents and the outside world to learn the optimal strategy that can maximize the reward. "Trial and error" is the core of reinforcement learning, and the purpose of reinforcement learning is to constantly obtain benefits from "trial and error". The reinforcement learning approach is applied to vehicular network users, so that vehicular network users can learn to obtain communication benefits in addition to communication. The users near the vehicular network collaborate in communication, reduce the waste of system resources, and at the same time enhance the communication rate of vehicles and the base station. In the multi-point cooperative transmission of vehicular networks, if the method of reinforcement learning can be used to make idle vehicles constantly learn to obtain benefits and improve the efficiency of information transmission, it is bound to promote the improvement of the communication ability of unmanned driving.

As pointed out above, vehicular networks can adopt multiple base stations for a user with services, which can easily interfere with other cells, occupy large channel resources,

and require high delay. So, the terminal vehicles begin to use continuous reinforcement learning to obtain their own benefits. In this paper, our innovation is to utilize the idle users of the surrounding vehicular networks to carry out self-learning through the Q-learning algorithm with the help of reinforcement learning and assist the vehicle users of the vehicular networks to carry out collaborative communication. This method not only can reduce the interference with other surrounding areas and occupy channel resources, but also can make full use of system resources and improve the overall performance of the vehicular networks. To boost the development of self-driving technology, the first thing to do is to improve the communication ability of the vehicular network coordination. Therefore, this topic adopts a benefit learning method to discuss how to effectively choose relay nodes, improve the performance of the coordination transmission system, and finally realize the vehicular network traffic capacity ascension.

## 2. System Model and Tool Analysis

### 2.1. System Model

In this paper, the system model of a vehicular network is built on the basis of a double link structure. As shown in Figure 1, there are macro base stations and micro base stations that provide services for vehicle-mounted users and transmit information at the same time. Macro base stations have wider communication coverage and can provide services for all users in the cell. However, its communication efficiency is usually low due to the influence of the system and communication frequency band. The problem of low communication efficiency can be solved by adding multiple micro base stations in the cell area covered by macro base stations. Therefore, the terminals have dual link capability, which can communicate with the macro and micro base stations at the same time for better service.

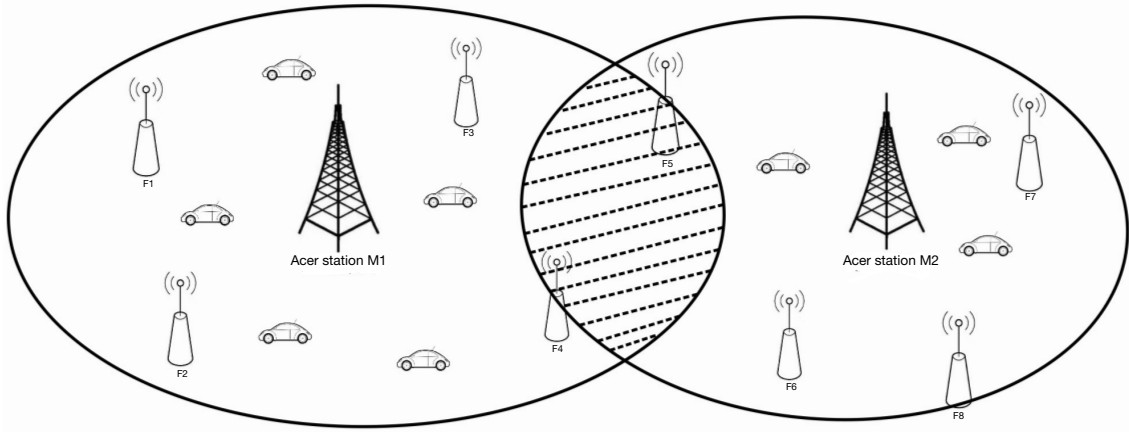

**Figure 1.** System model.

In order to ensure the quality of communication, more micro base stations are usually deployed in the cell. However, the micro base stations will interfere with each other and the communication quality of end users is reduced. In fact, the interference of micro base stations, especially those at the edge of the cell, mostly comes from the base station of the adjacent cell. The end users in the edge region are far away from the macro base station of their own cell and receive a weaker signal, so they are more susceptible to the interference of the z signal sent by the base station of the adjacent cell.

In order to solve the above problems and improve the communication service quality of the end users in the edge area, we need to obtain the signal strength and communication rate received by the end users. For end user $i$, the signal to interference plus noise ratio (*SINR*) [4] received from macro base station $g$ is expressed as follows:

$$SINR_i^{macro} = \frac{P_g \left| h_g^i \right|^2}{\sigma^2} \tag{1}$$

For the terminal covered by the micro base station, the signal to noise plus interference ratio of the micro base station received by the terminal is shown as follows:

$$SINR_i^{pico} = \frac{P_e \left| h_e^i \right|^2}{\sum_{j \neq e} P_j \left| h_e^j \right|^2 x_j + \sigma^2} \tag{2}$$

where $Pg$ represents the transmit power of macro base station $G$ located in the center of the cell, $Pe$ represents the transmit power of micro base station $E$, and $h_g{}^i$ and $h_e{}^i$ represent the path loss from the macro base station and the micro base station to terminal $I$, respectively. $\alpha^2$ represents the additive white Gaussian noise widely existing in the environment. For the dual-link network, due to the use of a large number of micro base station networks of the same system for network coverage, there is serious interference between the cells. $\sum_{j \neq e} P_j \left| h_e^i \right|^2 x_j$ is the interference of the neighboring cells to the local cell, which is also the main objective of our optimization. After the signal to noise plus interference ratio is obtained, Shannon's theorem is used to estimate the communication rate of the communication system in engineering. Shannon's theorem [5] is as follows:

$$R = W log_2(1 + \frac{S}{N}) \tag{3}$$

where $R$ is the communication rate, $W$ is the channel bandwidth, and $S/N$ is the noise ratio.

In the approximate estimation, we often use SINR instead of $S/N$ to estimate the channel performance. Therefore, the limit communication rate of terminal $I$ is estimated as follows.

$$\begin{aligned} R_i &= R^{macro} + R^{pico} \\ &= W^{macro} log_2(1 + SINR_i^{macro}) + W^{pico} log_2 \left(1 + SINR_i^{pico}\right) \end{aligned} \tag{4}$$

The communication rate of terminal $I$ consists of two parts. One part is the communication rate provided by the macro base station, and the other part is the communication rate provided by the micro base station. Based on the dual-connection technology, users can obtain services from different standard base stations at the same time, which greatly improves the quality of service of users and improves the utilization rate of base stations.

### 2.2. Multi-Point Cooperative Transmission Mechanism for Vehicular Users

In the vehicular network, the core idea of multi-point cooperative transmission is that the vehicular network users make use of other vehicular network users to cooperate when communicating with the base station. The multi-point cooperative transmission can transmit signals to the target which is unable to communicate normally due to serious interference or high path loss. It also can establish communication links in special occasions and improve communication efficiency by coordinating the information of multiple neighboring vehicular network users and carrying out multi-point cooperative transmission services through other neighboring vehicular network users. In the vehicular cooperative network, as shown in Figure 2, when some vehicular network users are in a state of suspended communication—that is, when they do not effectively communicate with the base station—other vehicular network users in the surrounding can use it to carry out cooperative communication services when communicating with the base station [6]. This technology not only occupies the original channel resources of other cells, but also has high technical requirements for capacity, delay, and signaling overhead. This paper studies the collaborative multi-point transmission for the vehicle network users by using surrounding suspended communication in particular, which can make full use of channel resources, improve the transmission rate of the normal communication vehicle network users, and minimize the interference to vehicular networks users in other areas. Therefore, the application of multi-point cooperative transmission technology in the communication between

users and base stations in such vehicular networks can improve the signal transmission efficiency and the overall performance of the system.

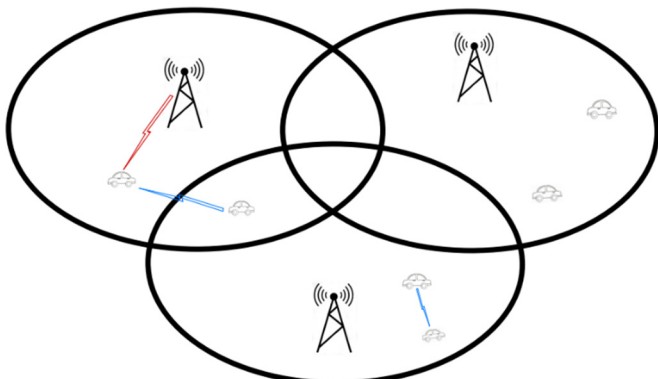

**Figure 2.** Vehicular network collaboration model.

*2.3. Reinforcement Learning of Q-Learning*

In reinforcement learning, the agent makes a decision and changes the state of the environment. The environment feeds the benefit back to the agent through the related algorithm so that the agent gradually learns the optimal decision. Among them, in *Q*-learning [7], the agent makes decisions through the *Q* matrix, which is a value matrix with the matrix table header of state and action. The agent checks its own state, selects actions according to the matrix values corresponding to all actions in the corresponding state column, and reacts to the environment. The environment feeds back the reward matrix and modifies the probability map of the corresponding action by changing the *Q* matrix in different states. The corresponding *Q* matrix update expression is as follows:

$$Q(s_t, a_t) = (1 - a)Q(s_t, a_t) + a(r_{t+1} + \gamma max_a Q(s_{t+1}, a)) \tag{5}$$

where *a* is the learning rate and $r_{t+1}$ is the reward matrix of the feedback from the environment. Noticing that *Q*-learning algorithm can apply for coordinated scheduling strategies in different scenarios and obtain the best balanced operation strategy [8], we utilize it here to achieve greater advantages in solving complex learning models.

**3. Joint Cell Anti-Jamming Map Based on Dual Connection**

*3.1. Interference Graph Model Based on Graph Theory*

As the network environment becomes more and more complex, it is difficult to complete the anti-interference task by simply using the traditional dual-link network technology. Therefore, it is necessary to use new anti-jamming mechanisms and related algorithms. In the traditional dual-link vehicular network technology model, the interference between cells is mainly from the signal transmitted by the base station of the adjacent cell. We can show the interference relationship of the cell cluster of the anti-interference algorithm implemented in the system model, as shown in Figure 3. In the joint interference graph based on graph theory, we first take the cell as the node of the graph and use edges to connect the nodes of neighboring cells with interference relations to form an undirected graph. Since the graph is planar, according to the four-color theorem [9], the maximum clique size of the graph is 4. In addition, in the conventional cellular network layout, if the base station can meet the maximum coverage area, the number of neighboring cells of each cell is 6; that is, in the conventional cellular network inter-cell interference diagram, the maximum cell size is 3.

In order to preferably solve the complex problem of the inter-cell interference algorithm, first, the graph is clustered [10]. As shown in Figure 3, pairwise adjacent base station nodes constitute the base station clique, and we need to conduct maximum clique searches on the cell interference diagram in Figure 3. For example, in Figure 3, nodes 4, 5, and

7 form a cluster of users and the cluster can independently execute the proportional fair scheduling algorithm, which schedules interval interference coordination [11] based on time domain segmentation. However, cluster {1, 4, 5} and cluster {4, 5, 7} will not be able to perform this anti-jamming algorithm at the same time. This is because the same two clusters of the launch base station {4, 5} cannot run two different scheduling policies at the same time.

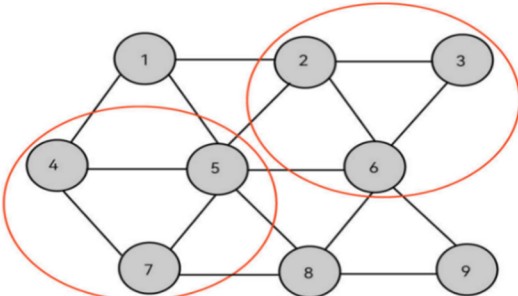

**Figure 3.** Intercell interference graph clustering.

In order to solve the conflict problem of the cluster, which is the scheduling strategy, we designed a conflict graph. In Figure 4, {A, B..., H} represents the cluster in Figure 3. For example, point *A* represents the {1, 4, 5} cluster in Figure 3, point *B* represents {1, 2, 5}, and so on. Moreover, points *A* and *B* are connected by an edge because they share nodes {1, 5}. If the nodes in Figure 3 are not connected, then the clusters in Figure 3 represented by these nodes can run their respective scheduling algorithms simultaneously. Figure 4 shows that the solution problem of this cell is a maximum clique problem. Through the maximum clique search problem, the maximum clique approximate search algorithm for the inter-cell interference graph of the cellular network and the heuristic search algorithm for the non-planar graph are given as follows.

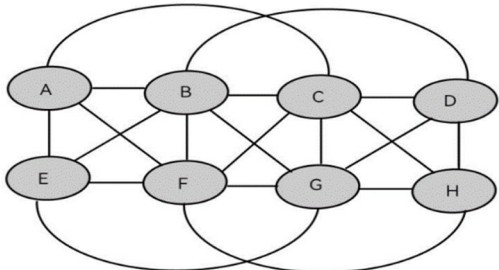

**Figure 4.** Cell cluster scheduling algorithm conflict graph.

### 3.2. Maximum Clique Search Algorithm for Inter-Cell Interference Graph

In Figure 3, the inter-cell interference graph constructed by the honeycomb structure is represented by a planar graph. We define the maximal clique here as the corresponding correlation graph if it is not contained by any other clique, i.e., it is not a proper subset of any other clique [12]. Because of its honeycomb structure, the size of the node groups connected pairwise in this kind of graph is at most 3. That is, the size of the maximal clique is at most 3 and each mesh represents a maximum clique. In other words, the corresponding maximal clique can be obtained by searching all the mesh in the graph.

When searching the maximum clique of the graph, which means searching the mesh, we first need to obtain the position coordinates of all the base stations and construct the interference graph *G(V,E)*. In graph*G*, node set *V* is composed of each base station, while edge set *E* is composed by connecting adjacent nodes with interference relations. Then, the above nodes are selected one by one as the origin of the plane rectangular coordinate system to form the plane rectangular coordinate system, and all nodes connected with the origin are added to node set *Cvi*. By calculating the anti-clockwise angle between the nodes



in *Cvi* and the horizontal coordinate axis, the corresponding included angle set *Avn* can be obtained. In this way, we can derive *Con(Cv1, Cv2... The Cvn)* and *Ang(Av1, Av2,...,Avn)*. By sorting *Con* based on *Ang*, we can obtain all maximal cliques of the interference graph. The process of the algorithm is detailed by Algorithm 1.

---

**Algorithm 1** Cluster Searching Algorithm for Interference Graph

---

1: **Input**: $G(Vn, Em)$, $Con(C_{V1}, C_{V2},...,C_{Vn})$, $Ang(A_{V1}, A_{V2}, ...,A_{Vn})$
2: **for** $i = 1$ to n **do**
3:     $K = \text{length}(C_{Vi})$
4:     **while** $K \geq 1$ **do**
5:         **for** $j = 1$ to $K$ **do**
6:             **if** $A_{Vi}(j) > A_{Vi}(j + 1)$ **then**
7:                 temp $= A_{Vi}(j + 1)$
8:                 $A_{Vi}(j + 1) = A_{Vi}(j)$
9:                 $A_{Vi}(j) = \text{temp}$
10:                 temp $= C_{Vi}(j + 1)$
11:                 $C_{Vi}(j + 1) = C_{Vi}(j)$
12:                 $C_{Vi}(j) = \text{temp}$
13:             **end if**
14:         **end for**
15:         $K = K - 1$
16:     **end while**
17: **end for**
18: **for** $i = 1$ to $n$ **do**
19:     **for** $j = 1$ to length($C_{Vi}$) **do**
20:         Put $\{V_i, V_j, V_{j+1}\}$ into *Cli*
21:     **end for**
22: **end for**
23: Delete the repeating items of the *Cli*
24: **Output**: *Cli*

---

### 3.3. Maximal Clique Approximate Search Algorithm

In order to resolve the problem that clusters with the same nodes can be simultaneously scheduled, we propose the heuristic search algorithm for the maximum clique approximate search, and the algorithm failure caused by the vehicular network not arranged according to the cellular network structure can be avoided when the complement graph of the conflict graph composed of the cluster set is searched for the maximal clique. Therefore, firstly, all nodes in the graph are traversed and all nodes connected to the visited nodes are put into the alternative point set *Cd*. The visited nodes are put into the point set *Cm*. After selecting the largest node in *Cd* and checking whether it is connected to all *Cm* nodes, we add it to *Cm* if so. If not, the node is not added and then deleted. We iterate over and over again until *Cd* is an empty set. The maximum cliques in the graph can be searched out as much as possible by first checking the node with the largest degree, and all nodes are traversed once to ensure the inter-cluster fairness of all cell clusters. The process of the algorithm is detailed by Algorithm 2.

By Algorithm 1, the cell cluster aggregation *Cli* under the interference graph is obtained in a finite time complexity. By Algorithm 2, we can obtain the maximum clique *Cm* of the scheduling conflict graph complement graph. Through *Cli* and *Cm* given in Algorithms 1 and 2, we can further explore how to schedule the base station to maximize the user's performance and maintain the scheduling fairness among users.

---

**Algorithm 2** Cluster Searching Algorithm for Conflict Graph

---

1:**Input:** $\overline{G}_{col}$ (*V,E*)
2:n = length(*V*)
3:*Cm* = Ø
4:**for** i = 1 to n **do**
5:　　*Cd* = *V*$_i$
6:　　$N_v$ = search_adjacent_node(*Vi*)
7:　　*K* = length(*Nv*)
8:　　**while** *Nv* != Ø **do**
9:　　　　Node = Search_max_degree($N_v$)
10:　　　　　　**if** cluster_judgement(*Node, Cd*) = =1 **then**
11:　　　　　　　　*C$_d$* = *C$_d$*∪ *Node*
12:　　　　　　**end if**
13:　　　　　　Delete *Node* from $N_V$
14:　　　**end while**
15:　　*C$_m$* = *C$_m$* ∪ *C$_d$*
16:Delete the repeat item in *C$_m$*
17:**Output**:*C$_m$*

---

*3.4. Interference Coordination Algorithm Based on Interference Graph*

In the interference coordination algorithm based on the interference graph, we use *X* to represent the sequence number of the conflicting clique, and then the effective communication rate of user *I* in clique *X* is as follows.

$$R_i^x = \sum_{l=1}^{t} p_i^x(l) r_i^x(l) \tag{6}$$

where *t* is the feasible scheduling permutation and combination in clique *x* and represents the communication rate that user *i* can achieve when scheduling in clique *x* with combination *l*.

In the interference coordination algorithm, for any user, the actual communication rate is composed of three parts. The first part is the rate achieved by the scheduling conflict clique when it uses the anti-jamming mechanism to schedule users. The second part is the rate achieved when the conflict clique does not use the interference coordination mechanism. The third part is the communication rate provided by macro base station. Therefore, for user *i*, the actual statistical rate is as follows.

$$R_i = \sum_{x=1}^{o} p_x R_i^x + \sum_{x=1}^{X-o} p_x p_i^{pico} r_i^{pico} + p_i^{macro} r_i^{macro} \tag{7}$$

where *X* represents the set of all maximal cliques in the complement graph of scheduling conflict graph and *o* represents the set of maximal cliques containing user *x* in *X*, which is a subset of *X*. Representing the scheduling ratio corresponding as clique*X* and the statistical communication rate of user *i* can be obtained when all maximal cliques in *o* are not scheduled.

Based on (7), we can express the statistical communication rate of all users as our optimization model as follows.

$$maximize \sum_{i=1}^{n} log f0 ( \sum_{x=1}^{o} p_x R_i^x + \sum_{x=1}^{X-o} p_x p_i^{pico} r_i^{pico} + p_i^{macro} r_i^{macro})$$
$$s.t. \sum_{x=1}^{X} p_x = 1$$
$$\sum_{i=1}^{n} p_i^{pico} = 1$$
$$\sum_{i=1}^{n} p_i^{macro} = 1$$
$$p_x > 0, \forall x$$
$$p_i^{macro} > 0, \forall i$$
$$p_i^{pico} > 0, \forall i$$

(8)

The system function in Equation (8) is a composite function, which includes both the scheduling ratio of the maximal clique in the complement graph of the conflict graph and the scheduling ratio of the permutation and combination of each scheduling in the corresponding maximal clique. Therefore, as long as (8) can be solved, the problem of obtaining the optimal network performance in a limited time and the problem of maximizing the performance of which cliques and which users can be served can be solved. The joint interference coordination mechanism can be obtained. In fact, (8) is an optimization problem under the constraints of equality. Its variable is the scheduling ratio of each conflict clique, and the scheduling ratio of different clique scheduling combinations is implied in $R_i^x$, which allows us to give the scheduling ratio of the conflict clique. Regarding the operation of priority scheduling mode pairs, we first give the priority iteration formula of the conflict clique:

$$p_x = \sum_{i=1}^{n} \frac{R_i^x}{R_i}$$

(9)

where $R_i^x$ is the cumulative average rate of user $i$ and $R_i$ is the cumulative average rate of user $i$ when scheduling conflict clique $x$. In order to enhance the comprehensive performance of communication system users, we select the highest priority conflict to implement group scheduling between fairness and overall performance through continuous iteration of (9). It can be solved when there is a group interference coordination-related operation scheduling conflict in the limited time complexity to achieve optimal network performance.

In order to solve the problem that users in a scheduling conflict clique should be served to maximize the system performance, we need to further expand (8) as follows to solve the scheduling combination:

$$maximize \sum_{i=1}^{n} log f0 ( \sum_{x=1}^{o} p_x \sum_{l=1}^{t} p_i^x(l) r_i^x(l) + \sum_{x=1}^{X-o} p_x p_i^{pico} r_i^{pico} + p_i^{macro} r_i^{macro})$$
$$s.t. \sum_{x=1}^{X} p_x = 1$$
$$\sum_{l=1}^{t} p_i^x(l) = 1, \forall x, i$$
$$\sum_{i=1}^{n} p_i^{pico} = 1$$
$$\sum_{i=1}^{n} p_i^{macro} = 1$$
$$p_x > 0, \forall x$$
$$p_i^x(l) > 0, \forall x, i, l$$
$$p_i^{macro} > 0, \forall i$$
$$p_i^{pico} > 0, \forall i$$

(10)

In (10), Equation (6) is used to expand in (8) to solve the scheduling combination in each maximal clique.

For scheduling combinations in any maximal clique, we use the same method to solve the priority iteration formula as the following section, and the corresponding priority scheduling mechanism can be obtained as

$$\rho_i^x = \sum_{t \in x} \frac{R_i^x(k)}{R_i(k-1)} \tag{11}$$

$$\rho_i^{macro} = \frac{r_i^{macro}(k)}{R_i(k-1)} \tag{12}$$

## 4. Collaborative Transmission Mechanism Based on Q-Learning for Vehicular Networks

### 4.1. Selection of Reinforcement Learning Algorithm

In reinforcement learning, there is always a quandary of exploitation dilemma [13]. Only by forming a compromise among explorations can we achieve a positive learning effect in practical applications. Among them, there are two commonly used algorithms for exploration. One of them is the greedy algorithm [14], in which the agent directly selects the action with the highest value in the corresponding state as the decision and the environment uses the reward matrix to feedback the *Q* matrix. This algorithm needs to set a certain exploration probability and the agent randomly selects the action to avoid falling into the local extreme value in the exploration state. The details can be found in Algorithm 3 as follows.

---
**Algorithm 3** Q-learning algorithm 1 ($\epsilon$-greedy)

---
1: Initialize *Q(s,a)* arbitrary
2: Repeat (for all episode):
3:  if random $<$ $P_{\text{investgate}}$
4:      Random take action *a*
5:   else:
6:      Take action a which is the greatest in *Q(s',a)*
7: $Q(s_t, a_t) = (1-a)Q(s_t, a_t) + a(r_{t+1} + \gamma max_a Q(s_{t+1}, a))$
8: Update s'

---

The other is the softmax exploration algorithm [15], in which the agent must normalize the probability of each action according to the softmax function. The details can be found in Algorithm 4 as follows.

---
**Algorithm 4** Q-learning algorithm 2 (softmax)

---
1: Initialize *Q(s,a)* arbitrary
2: Repeat (for all episode):
3 : $A = max_{ai} \frac{e^{Q(s',a_i)}}{\sum_{j=1}^{n} e^{Q(s',a_j)}}$
4: Take action a correspond with A
5:  $Q(s_t, a_t) = (1-a)Q(s_t, a_t) + a(r_{t+1} + \gamma max_a Q(s_{t+1}, a))$
6: Update s'

---

From the above, the greedy method is adopted to realize the collaboration between collaborative multi-point transmission and joint interference coordination mechanism in vehicular networks, and so we select it in this paper.

### 4.2. Multi-Point Collaborative Transmission with Revenue Learning

Multi-point cooperative transmission is a communication operation mechanism that uses other neighboring terminals to send signals to another terminal to improve the signal

to noise ratio of the target terminal. It was introduced earlier and will not be repeated here. Under the joint anti-interference modulation, some vehicular network users will be idle at some moments. Obviously, if these idle vehicular network users can be used to cooperate with the surrounding vehicular network users for communication, the overall performance of the vehicular network system will be further improved. However, since the vehicular network of the multi-point collaborative transmission mechanism needs to cooperate with the joint interference coordination mechanism, we use idle vehicular network users in the process of scheduling with the traditional arithmetic based on the utility function optimization, resulting in too many variables and a cumbersome process of calculation. Moreover, in the large-scale network, this will lead to computational resources and cost too much. Therefore, the method of revenue self-learning [16] is adopted in this paper to conduct scheduling among vehicular network users; that is, the vehicular network users can choose the optimal relay point for cooperation service through the revenue learning mechanism. This paper proposes a reinforcement learning algorithm of Q-learning, which is combined with the vehicular network of the joint anti-interference coordination algorithm. The vehicular network can continuously carry out self-learning and obtain profits to schedule cooperative multi-point transmission services and make full use of idle base stations to improve the overall performance of the system.

When using the reinforcement learning algorithm of Q-learning, we should fully understand what the states, actions, and rewards are, and fully understand the $Q$ matrix and payoff matrix of the learning scheduling strategy. Among them, the state represents the current connection status between each vehicular network user. The action represents the vehicular network user who will be taken to the next step to connect the user which has temporarily stopped communication by the vehicular network. That is, they cooperate to serve a vehicular network user. The reward is about the communication rate of the system communication. The $Q$ matrix is constructed by learning the scheduling strategy. The vehicular network user at a certain time slot may be communicating with the base station or not communicating temporarily. They can take actions such as connecting with the cell base station, stopping the communication and not cooperating to serve other users, cooperating to serve other users 1, cooperating to serve other users 2, etc. The state is the permutation and combination of the connection running in the current cell cluster. For each base station, its $Q$ matrix is a matrix of size ((number of users in the cell cluster +1) * Number of users in the cell cluster). For a cell cluster, the $Q$ matrix of a base station is shown in Table 1:

**Table 1.** Q matrix example of a base station in a cell cluster.

| Combination / Actions | User 1 | User 2 | User 3 ... | User *n* |
|---|---|---|---|---|
| Leave unused | | | | |
| User 1 | - | | | |
| User 2 | | - | | |
| User 3 | | | - | |
| ... | | | - | |
| User n | | | | - |

In the joint interference coordination mechanism, the optimization objective is the logarithm sum of the effective rates of each user. So, we use the logarithm sum of the effective rates of all users in the cell cluster as the value of the payoff matrix to keep consistent with the algorithm in the joint interference coordination mechanism. Considering the practical problems in statistical calculation, we use the logarithm sum of the cumulative average rate of each user in the cell cluster to construct the payoff matrix and $Q$ matrix; that is, we use $\sum_{i=1}^{m} ln f0 (R_i)$ as the feedback value of the payoff matrix, where $R_i$ is the communication rate of user $i$ in the cell cluster. The specific algorithm calculation process is shown in Algorithm 5.

---

**Algorithm 5** Collaborative multipoint transmission collaborative algorithm flow

---

1: Initialize $Q$(action, combination) for all the cluster arbitrary
2: Repeat (for all time_slots):
3:   Initiate process below for the cluster scheduled in this time slot:
4:   if random $<$ $P_{\text{investgate}}$
5:     Random take action for all the unused base station in the cluster
6:   else:
7:     Take action with the greatest value in $Q$ for all the stations
8:   Update $Q$ as below:
9: $Q(\text{action}_t, \text{combination}_i) = (1 - a)Q(\text{action}_t, \text{combination}_t)$

---

When scheduling the joint interference coordination mechanism, the idle base station updates the $Q$ matrix in each time slot. The specific Algorithm 5 is as follows.

1. Determine the coordinates of the small cell network and draw the interval interference diagram.

2. According to the obtained interval interference graph, conduct a maximum clique search of the interval interference graph to obtain a scheduling conflict graph. The corresponding learning matrix ($Q$ matrix) is initialized for all cell clusters.

3. The anti-joint interference algorithm is used to conduct the maximum clique approximate search on the complement graph of the scheduling conflict graph, and all combinations of maximum cliques are solved with priority, and the maximum clique with the highest priority is scheduled to execute the anti-interference scheduling algorithm

4. Calculate the priority of different scheduling combinations of the maximum clique executing the anti-jamming scheduling algorithm through (11) in the second largest point, and the combination with the highest scheduling priority. Use (12) of the second largest points to calculate the priority of each user to connect to the macro station, and schedule the link with the highest priority.

5. According to the $Q$ matrix of the scheduled cell cluster and the cooperative multi-point transmission cooperation algorithm, the cooperation decisions to be executed by all idle vehicular network users are determined.

6. Record the actual communication rate of each user in the time slot and update the revenue matrix.

7. Repeat 1–6.

The implementation of the above algorithm is based on the joint multi-point cooperative transmission and joint interference coordination mechanism under reinforcement learning. Algorithm 5 can effectively improve the call rate of vehicular network users and then the global communication performance.

## 5. Simulation Results and Comparative Analysis

In the simulation parameter setting, a square area with 200 m side length is assumed as the space of the communication simulation [17] and a square space with 50 m side length is assumed as the hotspot area which is covered by a high-density small cellular network. The user positions are randomly distributed in this space to simulate the real situation. Meanwhile, we use the cost-231 Walfisch–Ikegami model [18] as the channel attenuation model. The channel attenuation model is $L_{los}(\text{dB}) = 194.64 + 26log f0(d)$ for the macro base station and $L_0(\text{dB}) = 189.65 + 20log(d)$ for the micro base station and vehicular network users. Other parameters are shown in Table 2.

**Table 2.** Parameter settings.

| Parameter | The Values |
| --- | --- |
| Acer station transmit power | 46 dbm |
| Transmit power for micro base station and vehicular network users | 20 dbm |
| Number of Acer stations | 1 |
| Number of micro base stations | $4 \leq U \leq 8$ |
| Number of users | Twice the number of micro base stations |
| Small scale fading | A lognormal distribution with a mean of 3 dB |
| Acer station bandwidth | 20 Mhz |
| Bandwidth for micro base stations and vehicular network users | 100 Mhz |

The communication rate of the vehicular network system is the sum of the rate of the macro base station and the rate of the micro base station, represented by $\sum_{i=1}^{m} R_i = W^{macro} log_2 \left(1 + SINR_i^{macro}\right) + W^{pico} log_2 \left(1 + SINR_i^{pico}\right)$, where *m* represents the number of cell users. In the multi-point cooperative transmission mechanism, the communication rate of the system is the sum of the rate of the macro base station, the rate of the micro base station, and the cooperation rate of the vehicular network users, which is represented by $\sum_{i=1}^{m} R_i = W^{macro} log_2 \left(1 + SINR_i^{macro}\right) + W^{pico} log_2 \left(1 + SINR_i^{pico}\right) + W^{users} log_2 (1 + SINR_i^{users})$. In Figure 5, the throughput of different network sizes is compared with different algorithm strategies. It can be seen from the figure that the space complexity increases with the increase in network scale. With the increase in the complexity of the network scale, the throughput of the joint anti-interference mechanism system gradually widens the gap with the throughput of the traditional dual-link mechanism, and the joint anti-jamming algorithm we adopted makes the total rate of the system rise faster. However, under the mechanism of the joint anti-jamming algorithm, there will always be vehicular network users temporarily stopping communicating with the base station. In order to make full use of these idle vehicular network users, we propose a multi-point cooperative transmission mechanism algorithm based on revenue learning. According to Figure 5, on the basis of the joint anti-interference algorithm, the multi-point cooperative transmission mechanism based on reinforcement learning algorithm is added. According to the overall throughput of the system, the system throughput of the multi-point cooperative transmission mechanism based on reinforcement learning algorithm is greater than that based on the joint anti-interference algorithm.

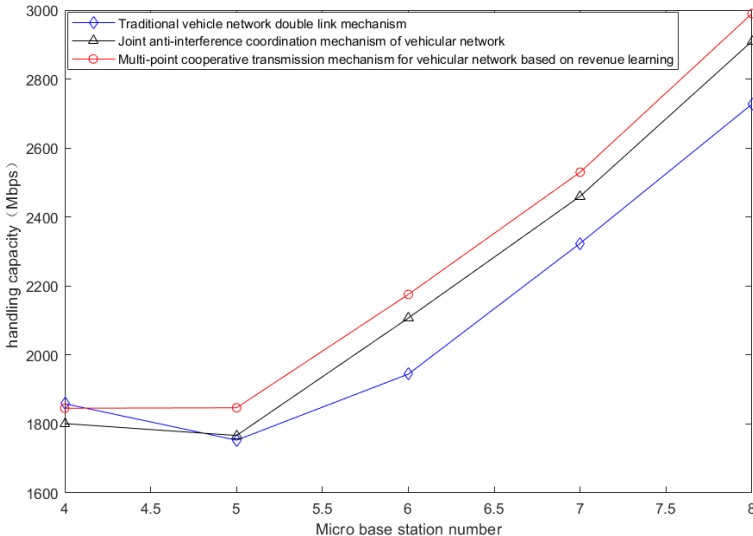

**Figure 5.** Comparison of system throughput under different policies.

In the vehicular network system, the average throughput of a cell is represented by $\left( \sum_{i=1}^{m} R_i = W^{macro} log_2 \left( 1 + SINR_i^{macro} \right) + W^{pico} log_2 \left( 1 + SINR_i^{pico} \right) \right) / N$, where N represents the total number of cell base stations. In the multi-point cooperative transmission mechanism, the average throughput of a cell is represented by $(\sum_{i=1}^{m} R_i = W^{macro} log_2 (1 + SINR_i^{macro}) + W^{pico} log_2 (1 + SINR_i^{pico}) + W^{users} log_2 (1 + SINR_i^{users}))) / N$. In Figure 6, with the increase in the number of cell users, the average throughput rate of the cell gradually decreases, which is caused by the increasing interference of the interval. Our proposed joint anti-jamming algorithm mechanism can effectively suppress the interference between cells, and the throughput decreases more slowly than the traditional double-link technology. In addition, the average throughput of the cells decreases more slowly than before under the multi-point cooperative transmission mechanism with reinforcement learning.

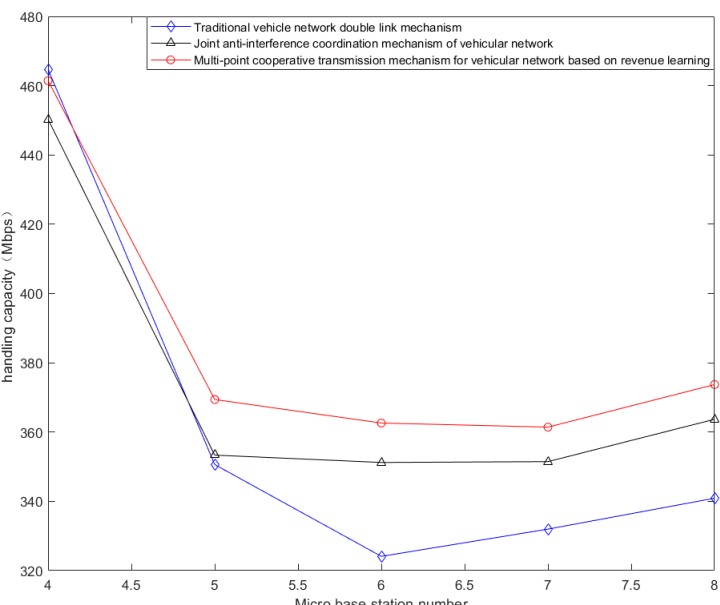

**Figure 6.** Comparison of cell average throughput under different strategies.

Figure 7 shows the overall system throughput improvement rate of vehicular network cooperative transmission mechanisms based on revenue learning and traditional vehicle network double connectivity, respectively. The throughput increasing rate of our vehicular network cooperative transmission mechanism is based on revenue learning and joint anti-interference mechanisms. We can clearly see that the system throughput growth rate of our collaborative transmission mechanism based on revenue learning is basically positive in terms of the number of base stations compared with the other two mechanisms. Therefore, from Figures 5–7, it is observed that the double link technique with a multi-point cooperative transmission mechanism based on reinforcement learning has obvious advantages compared with traditional joint and anti-jamming technology.

However, if only the throughput of the system is considered, it is easy to ignore the fairness of scheduling among users. In order to characterize the fairness of networks based on random topology [19], $\sum_{i=1}^{m} ln \, f0 \, (R_i)$ is used as a fairness index [20] to test the fairness differences of networks based on random topology when different scheduling policies are used, where $R_i$ is the average rate of users. Figure 8 shows the comparison of fairness coefficients under no policy. It can be seen from the display analysis of the data that the fairness of both the joint anti-interference coordination mechanism algorithm and the multi-point cooperative transmission mechanism under revenue learning is much better than that of the traditional double-link technology. Among them, although the use of multi-point cooperative transmission algorithm in scheduling the idle base station to send signals to other cells, the users will cause a certain degree of interference with the network. However,

after several rounds of learning, the $Q$ matrix can gradually converge to a better value so as to make a reasonable decision in multiple scheduling. It can be seen from Figure 8 that the multi-point cooperative transmission mechanism of a vehicular network based on revenue learning is better.

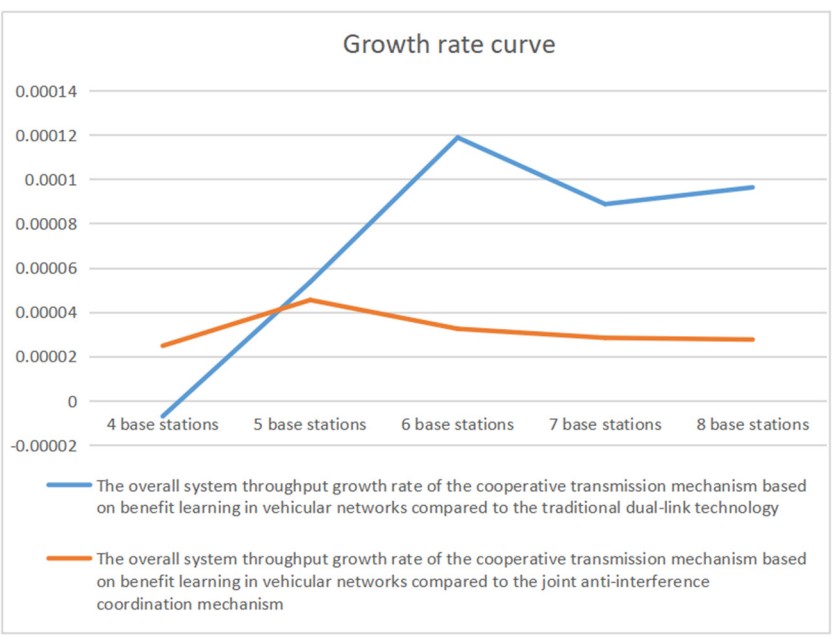

**Figure 7.** Growth rate curve.

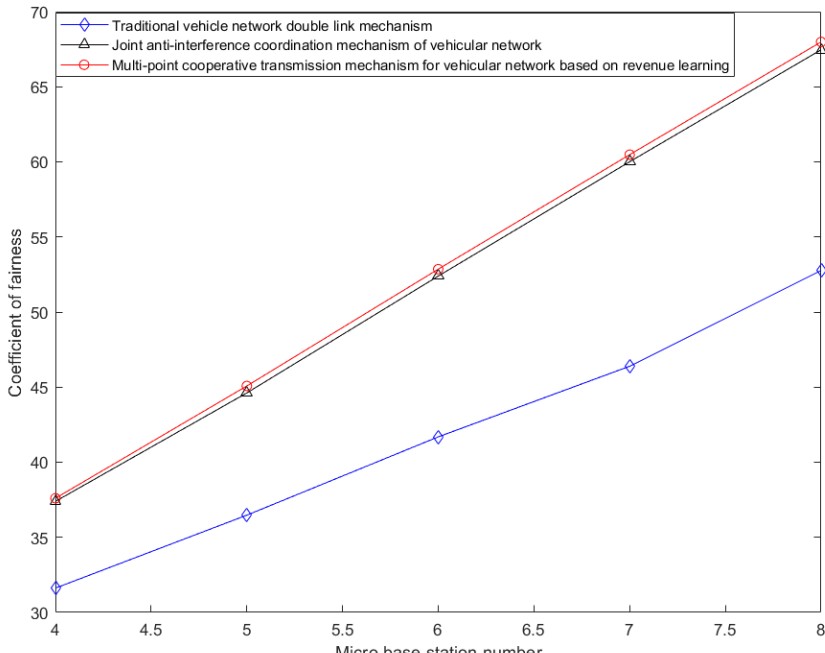

**Figure 8.** Comparison of fairness coefficients under no policy.

In brief, the simulation results show that the overall throughput of our proposed collaborative multi-point transmission mechanism has an average growth rate of 7.00% in different base stations compared with the traditional dual-link vehicular networks. Against the joint anti-jamming mechanism, the overall throughput of our proposed collaborative multi-point transmission mechanism has an average growth rate of 3.17% with different base stations.

## 6. Conclusions

In this paper, we have presented a cooperative transmission mechanism based on revenue learning with a kind of traditional double link technology for vehicular networks. Under the premise of combining algorithms for a joint interference mechanism and the income from the learning mechanism of multi-point cooperation transport, the proposed strategy improved the average throughput and fairness for the scheduling vehicular users. Simulation results show that our operative transmission mechanism based on revenue learning significantly obtains better performance compared to the conventional existing schemes, and the low-complexity suboptimal approaches can adequately balance the performance and complexity. Our study still has some limitations, e.g., the static models were randomly generated by the on-board network users. The dynamic revenue learning model of in-vehicular networks will be investigated in the future.

**Author Contributions:** Conceptualization, M.C. and M.N.; methodology, H.C., M.C., and Y.D.; software, Y.D. and M.C.; validation, Q.C. and S.Y.; formal analysis, Q.C.; investigation, S.Y.; resources, H.C.; data curation, M.C. and M.N.; writing—original draft preparation, Y.D., M.C., M.N., Q.C. and S.Y.; writing—review and editing, H.C. and M.C.; visualization, F.D.; supervision, F.D.; project administration, F.D.; funding acquisition, H.C. All authors have read and agreed to the published version of the manuscript.

**Funding:** This research was funded in part by the National Natural Science Foundation of China (grant numbers 61871433, 61828103), the South China Normal University National Undergraduate Innovation and Entrepreneurship Training Program (grant number S202210574115), and the Research Platform of South China Normal University and Foshan.

**Institutional Review Board Statement:** Not applicable.

**Informed Consent Statement:** Informed consent was obtained from all subjects involved in the study.

**Data Availability Statement:** Not applicable.

**Acknowledgments:** The authors would like to thank the supervisor, committee, and colleagues who provided help throughout this work and made this work possible. Furthermore, the authors would like to thank financial support from the National Natural Science Foundation of China and the South China Normal University.

**Conflicts of Interest:** The authors declare no conflict of interest.

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
