# Peer review of "Cooperative Transmission Mechanism Based on Revenue Learning for Vehicular Networks"

_applsci, doi:10.3390/app122412651_

Round 1

Reviewer 1 Report

In this paper, the authors propose clustering algorithms to mostly avoid interference and a reinforcement learning based algorithm to determine the actual transmission path, to increase the overall throughput of vehicular network. This topic is interesting, but many important explanations are missing. More comprehensive comparisons with the existing techniques need also be included. Finally, the simulation setup seems to be simplified, so at least a more detailed discussion related to more complex scenario is highly suggested. My detailed comments are as follows:

1.         In Sec. 1.2, the authors describe the necessity of multi-point cooperative transmission. However, there have been many existing works which propose to improve the efficiency of such scenarios. Please include state-of-the-art techniques and what are the major challenges given in these works. Finally, how the proposed method in this work can outperform the existing works?

2.         Sec. 1.3, why the authors specifically investigate and apply Q-learning in this work? If there are benefits that Q-learning can obtain for this specific application (vehicular network), please explain it. Also, what are the other reinforcement learning algorithms? Are other algorithms also applicable to be used in this application? How would the authors think of the overall performance for different RL algorithms?

3.         Sec 2., please explain what is the relationship between the algorithms proposed in 2.2 and 2.3. Is Algorithm 1 proposed to find all the possible interference graphs, while Algorithm 2 finds the optimal one from the candidates? Furthermore, please also explain what is the definition of “the maximum clique”?

4.         Sec. 3.1, why greedy algorithm is selected for the exploration-exploitation? It is obvious that greedy algorithm is highly possible to be trapped by local optimal, so how would the authors consider this problem? Would the result be different if more advanced algorithm is used, like simulated annealing, dynamic programming, etc.?

5.         Sec. 4, the setup for the simulation seems too simple. In specific, how are the number of stations and users are selected? Can the authors explain the complexity of such a setup, considering some practical scenarios? How would the authors think of the overall performance if the number of stations and users change? In addition, are the users/vehicles assumed to be static at certain positions or dynamically move around?

6.         Other comments:

l  Please correct the section numbers: there are two sections labeled with Sec. I.

l  Some equations with logarithms are not well presented with unrecognized symbols.

Reviewer 2 Report

This paper presents a study based on earnings learning with vehicular network multi-point collaborative transmission mechanism to allow better resource utilization of the double connection vehicular network. In this article, the vehicular network users communicate with the surrounding collaborative transmission. It used a Q-learning algorithm in the reinforcement learning process to enable vehicular network users to learn from each other and make cooperative decisions in different environments.

Overall:

1.       The article presented an excellent study of vehicular networks.

2.       The paper showed well-organized with proper structure, and the methodology and results are communicated.

This paper can be accepted. However, there are some points that the authors can handle:

1.       You have to highlight the contribution of your work and talk about the limitation of your work.

2.       You mentioned in the abstract, "The experimentation results show that our proposed approach performs to the optimum with our revenue machine learning model". It is better to give a percentage to show how is your work better than the other work.

3.       It would be better to add a section for your future work.

4.       You have to update the references because most of them are from before 2011, which is very old.
